# The Impact of the *No Jab No Play* and *No Jab No Pay* Legislation in Australia: A Scoping Review

**DOI:** 10.3390/ijerph20136219

**Published:** 2023-06-24

**Authors:** Sharyn Burns, Ranila Bhoyroo, Justine E. Leavy, Linda Portsmouth, Lynne Millar, Jonine Jancey, Jacqueline Hendriks, Hanna Saltis, Jenny Tohotoa, Christina Pollard

**Affiliations:** 1Collaboration for Evidence, Research and Impact in Public Health, School of Population Health, Curtin University, Bentley, WA 6102, Australia; 2Telethon Kids Institute, Perth, WA 6102, Australia

**Keywords:** childhood vaccination, vaccine hesitancy, parent attitudes and beliefs, childhood immunisation, *No Jab No Pay*, *No Jab No Play*, immunisation mandates, immunisation coverage, vaccination policy, financial sanctions

## Abstract

Australia has a long history of population-based immunisation programs including legislations. This paper reports on a review of evaluations of the impact of the federal No Jab No Pay (NJNPay) and state implemented No Jab No Play (NJNPlay) legislations on childhood immunisation coverage and related parental attitudes. Five databases were searched for peer-review papers (Medline (Ovid); Scopus; PsycInfo; ProQuest; and CINAHL). Additional searches were conducted in Google Scholar and Informit (Australian databases) for grey literature. Studies were included if they evaluated the impact of the Australian NJNPay and/or NJNPlay legislations. Ten evaluations were included: nine peer-review studies and one government report. Two studies specifically evaluated NJNPlay, five evaluated NJNPay, and three evaluated both legislations. Findings show small but gradual and significant increases in full coverage and increases in catch-up vaccination after the implementation of the legislations. Full coverage was lowest for lower and higher socio-economic groups. Mandates are influential in encouraging vaccination; however, inequities may exist for lower income families who are reliant on financial incentives and the need to enrol their children in early childhood centres. Vaccine refusal and hesitancy was more evident among higher income parents while practical barriers were more likely to impact lower income families. Interventions to address access and vaccine hesitancy will support these legislations.

## 1. Introduction

Immunisation is considered one of the most successful and cost-effective public health interventions to prevent infectious disease and protect public health and well-being [1]. Globally, childhood immunisation programs have led to a decline in disease transmission rates [2] and have provided the advantage of community-level protection or herd immunity in addition to individual level protection [3]. While immunisation coverage rates required to achieve herd immunity vary for individual diseases, Australia has set a target of 95% full coverage for childhood immunisation based on evidence, consultation, and practical considerations [3]. 

It is estimated the measles vaccine alone was responsible for averting approximately 23 million deaths globally between 2010 and 2018 [4]. Despite these advances, in recent years, outbreaks of childhood diseases have occurred in high-income countries, such as France, Ireland, and the US [5]. While significant progress towards global vaccination targets have been made, regional elimination targets have not been achieved [6]. Reductions in public compliance with vaccination mandates; sustained transmission in adjacent regions; and low immunisation rates in specific communities have been identified as threats for infectious disease prevention [7]. In Australia, the quarterly report for December 2022 for fully immunised (as per the National Immunisation Plan) children shows immunisation rates to be high, at 93.75%; 91.98%; and 94.27% for 1-, 2-, and 5-year-olds, respectively [8]. However, despite high overall childhood vaccination rates, these rates do not meet the 95% target [8]. Further, specific geographical areas and population groups within the country have lower childhood vaccination coverage, increasing the risks of local outbreaks [9,10]. 

Partial and non-vaccination are associated with issues around choice, acceptance, or reduced access [5]. While definitions of vaccine decision-making vary, it is acknowledged that vaccine decision-making falls on a continuum from acceptance to refusal. Vaccine hesitancy falls along the continuum and may include parents who have some concerns about some or all vaccinations. They may also delay or accept vaccination but be uncertain about their actions [11,12]. The perceptions, values, and beliefs of vaccine-hesitant and refusing parents influence their choice to vaccinate their child, with some making decisions based on their perception of the efficacy or safety of vaccines while others may be influenced by religious, philosophical, or political beliefs [13]. Access issues include availability, affordability, and accessibility of services, with suggestions that parental complacency may also contribute [5]. In Australia, lower rates of fully immunised children can be found at both ends of the socio-economic spectrum [14], with researchers suggesting partial or non-vaccinating families from higher socio-economic areas are more likely to be vaccine-hesitant or vaccine refusers, while those from lower socio-economic groups are more likely to experience issues of access [5,15,16].

While infant and early childhood vaccination programs have been available since the 1940s in Australia, the launch of the ‘Immunise Australia: Seven Point Plan’ (the Plan) in 1997 [16,17,18] provided the political impetus and strategic direction to increase childhood immunisation beyond the 53% full coverage at that time [19]. The Plan included incentives for parents and general practitioners; monitoring and evaluation targets; education and research; a measles elimination strategy; school entry requirements; and immunisation days [17]. Since then, a range of policies and incentives have been introduced at both state and national levels to strengthen coverage, including financial incentives [20]. Initially, all families received some financial incentives; however, in 2012, the financial incentive became means-tested [20]. 

Building on the Plan, in January 2016, the *No Jab No Pay* (NJNPay) legislation was introduced by the Australian Federal Government, linking fiscal incentives with child immunisation. This legislation abolished the option of conscientious objection for philosophical or religious reasons. Parents of fully immunised children (up to the age of 19) or children on an approved catch-up schedule or with approved medical exemptions are eligible for the payments linked to the Child Care Benefit, Child Care Rebate, and Family-Tax-Benefit-A [16,21,22]. 

In conjunction with NJNPay, the *No Jab No Play* (NJNPlay) legislation mandated varying conditions and restrictions regarding eligibility to attend early childhood education centres. Unlike NJNPay, this legislation is not implemented federally but by state and territory governments. Implementation has not been universal, with states and territories mandating legislation on different dates and with differing requirements [23]. Currently, five Australian states have NJNPlay policies, with varying provisions [22]. New South Wales (NSW) was the first state to implement a policy in 2014, including a conscientious objection clause. However, the option for this objection was removed in 2018. Victoria and Queensland implemented NJNPlay in 2016 [16], while Western Australia (WA) and South Australia (SA) implemented legislation in 2019 [23]. In general, NJNPlay requires children to be fully immunised to enrol in an early childhood education centre (childcare or kindergarten); however, there are a range of exemptions. For example, in WA, this refers to any child classified under regulation 10AB of the Public Health Regulations 2017 which includes children under a protection order or living in crisis accommodation [24]. 

Childhood vaccination rates have been steadily rising in Australia for several decades causing some researchers to question the need for mandatory vaccination and the true impact of these legislations [3,13,25]. While the initial intention of this review was to specifically explore the impact of NJNPlay, most studies evaluated both NJNPay and NJNPlay legislations. The lack of individual evaluation of NJNPlay, complicated by the timing of state-based implementation [23] required inclusion of both legislations. Therefore, this review explores the impact of the Australian federal NJNPay and state NJNPlay legislation on childhood vaccination coverage and the impact of the legislations on motivating parents to vaccinate their children. 

## 2. Materials and Methods

A scoping review was conducted in accordance with the Joanna Briggs Institute (JBI) methodology for scoping reviews [26] and reported in accordance with the Preferred Reporting Items for Systematic Review and Meta-Analysis Protocols Extension for Scoping Reviews (PRISMA-ScR) (https://prisma-statement.org/Extensions/ScopingReviews) (accessed on 17 November 2021). This review explored the broad question, ‘What is the impact of the No Jab No Pay and No Jab No Play legislations on childhood immunisation in Australia?’.

### Search Strategy

A preliminary search for existing scoping reviews on the topic was conducted in PROSPERO and JBI Database of Systematic Reviews and Implementation Reports to identify similar publications on the topic and avoid duplication. A specialist health-sciences librarian assisted in the design of the search strategy to locate peer-reviewed publications. Five databases were searched—Medline (Ovid), Scopus, PsycInfo, ProQuest, and CINAHL. Following this, additional searches were conducted in Google Scholar and Informit to identify any additional publications and to explore the grey literature. Texts were excluded from the review if they were secondary reviews of the literature and if they reported only anecdotal evidence. The final search was conducted on 1 December 2022. The search terms in Table 1 were adapted for the different databases. 

This review considered texts including primary studies (quantitative or qualitative) and grey literature from 2014 onwards. Papers were included if they evaluated the implementation of the Australian NJNPay and/or NJNPlay legislations; were peer-reviewed or were government reports; and were in the English language. Observational studies, commentaries, case studies, book chapters, dissertations, conference proceedings, and reviews were excluded. All references were downloaded into the EndNote 20.0 program. After removing duplicates, the references were imported onto Rayyan online platform for screening abstracts and titles (http://rayyan.qcri.org/) (accessed on 21 November 2021). The final list of full-text papers was confirmed by two authors (RB and SB). Reference lists of included publications were searched to identify any additional publications.

Data were extracted into a Microsoft Excel spreadsheet. Data extracted included study title, name of authors, year of publication, aim of study, study design, participant groups and methods, demographic information, study inclusion and exclusion criteria, data analysis, impact of the legislations on childhood immunisation coverage and catch-up rates, and parent attitudes and perceptions of the legislations.

## 3. Results

A total of 795 records were obtained from six databases from which 704 were retained after removing duplicates. The full selection process is outlined in Figure 1. 

### 3.1. Study Characteristics

Nine peer-reviewed studies [9,16,27,28,29,30,31,32,33] and one government report [15] were included in the review. Studies evaluated NJNPlay (*n* = 2) [15,33]; NJNPay (*n* = 5) [9,29,30,31,32]; and both legislations (*n* = 3) [16,27,28]. Four studies analysed national Australian Immunisation Register (AIR) data [16,28,31,33]. Two national studies collected quantitative data from parents [30,32]. Two studies were conducted in Victoria: a study in Melbourne used AIR data and data collected from parents/guardians and physicians [27]; and the second, a Government report, analysed AIR data and collected data from parents and service providers from metropolitan and rural Victoria [15]. Two studies were conducted in NSW: one analysed retrospective data from a single rural medical practice [29]; and one collected qualitative data from parents in rural NSW [9] (see Appendix A).

### 3.2. Impact of the Legislation on Childhood Immunisation

Overall, all studies which specifically included questions around the impact of the NJNPay/NJNPlay legislation reported positive outcomes. Frawley et al.’s (2018) national study of parents (*n* = 429) included a focus on the effect of the NJNPay legislation on parents’ intention to immunise their children [30]. Most parents (93.5%) reported that their youngest child’s vaccination was up-to-date. For some parents (2.6%), the NJNPay legislation prompted vaccination. However, others (1.2%) suggested that the legislation made them less likely to consider vaccination. Parents who had consulted a paediatrician during the previous year were more likely to report their child’s vaccination was up-to-date than those who had visited a complementary medicine practitioner; the latter group of parents were also less likely to report their child was vaccinated [30]. 

A retrospective clinical audit within a single medical practice in rural NSW assessed the incidence of catch-up vaccinations for children and adolescents ≤ 19 years and older than four years before and two years after the NJNPay policy implementation [29]. Prior to the legislation implementation (between 2012 and 2015), catch-up vaccination incident rates were 6.1%; these rates increased to 9.2% in 2016 and 7.8% in 2017 post-legislation implementation [29]. The evaluation of NJNPlay in Victoria found a 3.3% to 3.7% increase in immunisation coverage during the period 2016–2020 for 1-, 2-, and 5-year-olds, resulting in 95.4%, 93.1%, and 96.2% coverage for each age cohort, respectively [15]. 

The four studies analysing AIR data found small but significant increases in fully vaccinated children [16,28,31,33]. Attwell and colleagues’ (2020) time-series analyses (2009 to 2017) found that prior to mandates, full coverage had improved across all states between 2009 and 2014 for 1- and 5-year-old children. The positive trend continued after 2014 when NSW and then other states began to implement mandates. NSW, the first state to implement NJNPlay legislation, reported an annual increase of 1.25% for full coverage across age groups [28]. No significant difference in the impact of the policies between communities with high, medium, and low numbers of registered vaccine refusers were noted. The findings could not conclusively determine the specific impact of the federal NJNPay or the state/territory NJNPlay policies for most states or territories. However, the authors suggested that these policies are likely to have reinforced existing strategies [28]. 

Hull et al. (2020) used cross-sectional AIR data to determine the initial impact of the NJNPay legislation for children aged five to less-than-seven years, children aged seven to <10 years, and young people aged 10 to <20 years between December 2015 and 2017 (NJNPay period only) [31]. Greater increases in vaccination rates were observed for Aboriginal and Torres Strait Islander children and those with the lowest socio-economic status. However, catch-up rates were lower in remote areas [31]. The Victorian state-wide study found immunisation rates increased for Aboriginal and Torres Strait Islander children across all three age groups since 2016 with 2020 rates increasing to 96.8% and 97.9% for 1- and 5-year-old children, respectively, exceeding the 95% target level [15]. Across all three age groups, a higher proportion of Aboriginal and Torres Strait Islander children were vaccinated post-introduction of the legislation. Ten of fifteen LGAs in low Socio-Economic Indexes for Areas (SEIFA) areas exceeded the 95% full coverage benchmark compared to eight of fifteen high socio-economic areas. Access was suggested as the most likely reason for lower coverage in lower socio-economic areas. Parents reported that the NJNPay legislation was more likely to influence decisions to vaccinate than the NJNPlay due to financial incentives associated with NJNPay [15]. 

An analysis of national AIR data between 2014 and 2018 for children aged ≤5-years found increased vaccine coverage of 2% to 4% for 1-year-old children across all states and territories [16]. The authors suggested that NJNPay and more stringent documentation requirements implemented in 2016 resulted in a 2.7% and 3.8% increase in fully immunised 1-year-olds in 2017 and 2018 in WA. Increases in fully vaccinated 2-year-olds also varied across states and territories, with indications that NJNPay increased coverage in NSW by 1.2% in 2017 and, along with NJNPlay, by around 1.8% in 2018. Among 5-year-olds, NJNPay was associated with increases of 2–3% in WA and SA during this period. Full vaccination coverage improved after the implementation of NJNPay, with more significant increases in coverage among States that had implemented NJNPlay. Improvement in coverage was significantly higher among areas with greater socio-economic disadvantage, lower median income, greater dependence on benefits, and a higher pre-legislation baseline. Socio-economically advantaged areas with a lower baseline coverage were less responsive to NJNPay and NJNPlay legislation changes [16].

Toll and Li (2022) investigated the impact of NJNPlay using AIR data from January 2016 to December 2019 linked to regional characteristics from the Australian Bureau of Statistics. Analysis considered different implementation times of NJNPlay across states and territories allowing for intervention-and-control and before-and-after study design. These data demonstrated small but significant increases of around 1% across all age groups (1-, 2-, and 5-year-olds) after implementation of NJNPlay. Increases were greater for 2- and 5-year-old cohorts; however, this may be associated with a higher proportion of these age groups enrolling in childcare. The smallest policy effects were found in the highest socio-economic quartiles. In addition, some lower socio-economic quartiles reported insignificant policy effects [33]. 

### 3.3. Impact of the Legislation on Motivating Parents to Vaccinate 

The NJNPay and NJNPlay policies were found to significantly motivate vaccine-hesitant parents (16%) to attend hospital-provided clinics to discuss or catch-up on vaccines [27]. NJNPlay was slightly more influential in motivating attendance at clinics (8–9%) compared to NJNPay (5–6%). Most vaccine-accepting parents of under-vaccinated children (89%) planned to catch-up their child’s immunisation to enrol them in an early childhood education centre, compared to approximately half of vaccine-hesitant parents. While immunisation rates in this selective cohort did increase initially, children of vaccine-hesitant parents and those seeking a medical exemption were less likely to be fully immunised at seven months post attendance [27]. The Victorian study of 440 parents found that vaccine-hesitant and vaccine-refusing parents reported feeling discriminated against and socially isolated due to the legislation. Some parents also expressed feeling coerced and economically disadvantaged due to loss of financial benefits and expressed their belief that the legislation was an infringement on children’s rights to access early childhood education [15].

Non-vaccinating parents (*n* = 31) from northern regional NSW (Bryon Region) participated in interviews to explore the impacts of the NJNPay legislation [9]. Parents reported that their decision not to vaccinate was associated with an interest in health and wellbeing and promoting positive health. While participants discussed social conscientiousness and responsibility, they felt herd immunity was an overly simplistic concept. Most parents (*n* = 26) indicated that the legislation had minimal financial impact as their income precluded them accessing financial payments; they had made a choice not to receive payments for which they were eligible; and/or they did not use or rarely used childcare services. Participants suggested that mandates would not influence them to vaccinate their children and suggested alternative strategies to avoid vaccination such as increasing extended family support; considering informal childcare arrangements such as house sharing; reducing work and study commitments; and withdrawing their children from childcare services. Some expressed fear about further punitive measures for those declining vaccination highlighting the coercive nature of the financial incentives. Homeschooling or accessing education outside the mainstream system were also reported as solutions [9].

## 4. Discussion

The aim of this scoping review was to explore the impacts of the federal NJNPay (2016) and the state administered NJNPlay legislations on childhood immunisation coverage and parents’ attitudes to the legislations. Increases in full coverage of children ≤ 5-years in Australia after the federal implementation of NJNPay in 2016 [16,28,29,30,31] and state implementation of NJNPlay from 2014 onwards were noted [15,27,33]. It is evident that the sustained and comprehensive immunisation plan in Australia since 1998 has resulted in a significant increase in full coverage of 5-year-olds from 53% in 1995 [19] to 94.27% in 2022 [8], with mandates contributing to recent improvements. 

For example, in Victoria increases in full coverage between 2015 (prior to NJNPay and NJNPlay implementation in 2016) and 2020 for the 1-, 2-, and 5-year-old cohorts were statistically significant, with Victoria being the only jurisdiction to exceed 95% immunisation coverage for 5-year-olds in 2019 [15]. A national study which specifically analysed the impact of NJNPlay by region using intervention-control and before-and-after analysis found small but significant improvements in states that had implemented NJNPlay [33]. Others found NJNPay to encourage increased uptake of catch-up vaccinations [29,30,31].

The findings of this review highlight disparities between socio-economic groups which have been identified elsewhere in the literature [34]. Children at the lower and higher ends of the socio-economic continuum report the lowest coverage rates [8,14,16,33]. However, when considering the impact of the legislations, increases in vaccine coverage [15,16,32] and responses to catch-up vaccination programs [31] were found to be greater among lower socio-economic families. Parents indicated that the NJNPay legislation was more likely to influence their decisions to vaccinate their children if they were concerned about the potential loss of financial incentives [32], and analysis of longitudinal data found NJNPlay to be influential in encouraging vaccination for lower socio-economic parents [33]. Lower income partial or non-vaccinating families have also been found to be disproportionally impacted by NJNPlay, delaying parents’ opportunities for return to work and depriving children of educational opportunities [35]. However, while the implementation of NJNPlay varies across Australia, there are a variety of exemptions. For example, children in WA can be enrolled if they are on an approved catch-up scheme. Special consideration is afforded to a range of socio-economically disadvantaged groups and families experiencing hardship, including homelessness, domestic and family violence, displacement due to natural disasters, and those on humanitarian visas [24]. 

Vaccine refusal and vaccine hesitancy have been suggested as a key reason for non-vaccination among higher socio-economic families [5,10] with partial immunisation among some families highlighting their concerns about specific vaccinations [25]. Non-vaccinating higher-income parents from Byron Region, NSW, were less concerned about the financial impact of the policies [9], supporting findings that full coverage was lower in higher socio-economic quartiles [33]. Similarly, a qualitative study in high-income areas of Perth, WA, found that some participants identified specific vaccinations they would not consider [10]. While parents in the Perth study considered themselves well-educated and researched the ingredients and potential effects of vaccinations, they were influenced by high-profile anti-vaccine advocates [10]. Similar to the Byron Region study included in this review [9], parents in the Perth study [10] and another Perth study of vaccine-refusing parents in Fremantle, also a higher socio-economic area with low comparative coverage rates [36], found refusal or partial vaccination to be associated with distrust in governments, health professionals, and pharmaceutical companies and a belief that where they lived protected them and their children from specific diseases [10,36]. These findings highlight concerns that mandatory policies reliant on financial benefits may afford higher income parents’ opportunity to refuse vaccination, compared to families who may rely on the financial benefits associated with vaccination, whether it be in the form of incentives or employment opportunities afforded by access to childcare [3,33,35]. While vaccine refusers are of concern, it is recognised that vaccine-hesitant families are more amenable to interventions [3].

Conversely, partial or non-vaccination among lower socio-economic groups is suggested to be influenced by barriers to access associated with difficulty and cost accessing health care services (for example, transport, time off work, consultation fees) [34]. The findings of this review, along with comparisons of similar mandates in other high-income countries [5,28,37], highlight the need to target access, especially among lower socio-economic and migrant groups. 

Some population groups reported lower full coverage. In Victoria, the complexities associated with immunisation catch-up for children born overseas was noted [15]. Issues associated with updating AIR records for migrant and refugee children were also recognised [38]. These children and their families may be placed at unnecessary risk. For example, under-vaccinated young adults of migrant origin were found to be a high-risk group during the 2012 measles epidemic in Australia [39]. Targeted interventions are required to ensure that specific migrant and refugee groups are supported to access immunisation, and that strategies implemented to ensure their records are accessible. 

Encouragingly, some population groups exceeded targets. For example, full coverage for 5-year-old Aboriginal and Torres Strait Islander children exceeds the national target of 95% (96.09% at December 2022) and is higher than for non-Indigenous children [8]. In Victoria, full coverage of 5-year-old Aboriginal and Torres Strait Islander children was 97.9% post-implementation of NJNPlay [15]. However, despite these encouraging findings, some geographical areas report lower coverage for Aboriginal and Torres Strait Islander children [40]

While it is evident that the combined NJNPay and NJNPlay legislations have contributed to increased vaccination coverage among children, not all agree that mandates are necessary in a country with high overall coverage [3]. Debate about the necessity of mandates in countries with relatively high coverage centres largely around inequities associated with lack of access to early education and opportunity for parents to secure employment, both of which are more likely to impact lower socio-economic families [3,41]. Public health policy should aim for the least restrictive measures and avoid inequity by exacerbating socio-economic and educational disadvantage [41]. Such debate recognises the importance of mandates; however, it calls for greater consideration around the medical exemption process, noting the ethical and emotional pressure this process may place on physicians [41,42].

Similar vaccine mandates have been implemented in varying degrees in other high-income countries [5,37]. Political-cultural context along with vaccine policy history have informed decisions about mandates, with restrictions, incentives, and enforcements varying. For example, in the US, all states require vaccination to attend day-care or school. However, most allow non-medical exemptions. Unlike countries such as the US and France where polices impact school years, Australian NJNPlay legislation mandates only apply to early childhood education (childcare and kindergarten) enrolment [5]. A comparison of childhood vaccine ‘mandates’ across 149 countries found considerable variation in policies ranging from stringent mandates to recommended policies. The study concluded that the effectiveness of policies were context specific and influenced by political, social, and scientific drivers [37]. 

Evaluations included in this review collected data immediately before and after implementation of the NJNPay and NJNPlay legislations. Since the implementation of these legislations, the global coronavirus pandemic (COVID-19) has generated significant discussion about vaccination and immunisation programs, including the implementation of broader population level regulations and mandates [42]. This heightened awareness may impact parents’ decision to vaccinate their children which may motivate or dissuade intentions to vaccinate. 

## 5. Conclusions

The findings of this review demonstrate continued small improvements in childhood vaccination coverage in Australia since the implementation of the Immunisation Plan in 1997, with additional increases in full coverage and catch-up after the implementation of NJNPay federally and NJNPlay in some states. However, higher and lower socio-economic groups reported lowest levels of full coverage. Catch-up programs demonstrated significant increases in full coverage among lower socio-economic groups, highlighting the need for targeted interventions to enhance access and support for these parents. Higher income parents were more likely to discuss vaccine hesitancy and refusal. Strategies to address vaccine hesitancy are recommended. Continued improvements to ensure physical barriers to access are removed for lower income parents are required. Specific strategies to support migrants and refugees are also warranted. The varied implementation of NJNPlay across jurisdictions makes it difficult to discern the effectiveness of this legislation, although two studies did demonstrate small but significant improvement in full coverage as an impact of the legislation. The lack of specific evaluation of the impact of the NJNPlay legislation highlights the need for further evaluation. 

## Figures and Tables

**Figure 1 ijerph-20-06219-f001:**
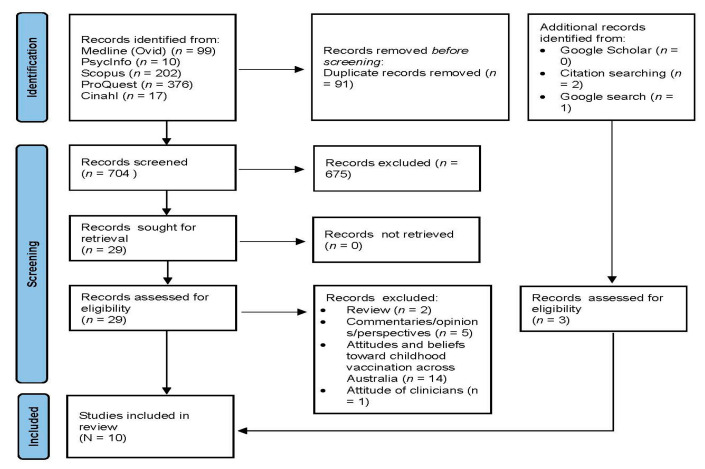
PRISMA diagram showing article and study selection.

**Table 1 ijerph-20-06219-t001:** Study selection and data extraction.

Concept 1	Concept 2	Concept 3
immuni* or vaccin* ADJ3 (polic* or legislation or law or mandate) or “no jab no play” or “no jab no pay” or NJNP	child or toddler or infant or baby or babies or paediatric or pediatric or parent	australia or victoria or new south wales or queensland or northern territory or tasmania or canberra or sydney or perth or melbourne or brisbane or hobart or darwin or adelaide

## Data Availability

Data are available upon reasonable request from the first author.

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
