# Peer review of "The Impact of the No Jab No Play and No Jab No Pay Legislation in Australia: A Scoping Review"

_ijerph, 2023, doi:10.3390/ijerph20136219_

Round 1

Reviewer 1 Report

Thank you for the opportunity to review this paper.

This paper reports on a review of evaluations of the impact of the federal No Jab No Pay (NJNPay) and state implemented No Jab No Play (NJNPlay) legislations on childhood immunisation coverage and related parental attitudes.

A very well written paper on an important topic, this is a valuable contribution to the knowledge base.

In abstract, it would be helpful to clearly indicate some key factors contributing to low coverage along the social gradient, not only for the higher socio-economic group (which are mentioned and are primarily motivational). This (access and practical barriers for lower socio-economic groups) is mentioned in discussion, and should be included in the abstract as well.  

Authors conclude that evaluation of mandates demonstrates continued improvements in childhood vaccination coverage. While in principle this is a correct statement, it would be helpful to include qualifiers such as “small improvements”.

There is a lack of discussion about the appropriateness of mandates (as a strategy to increase vaccination uptake), i.e.  are mandates helping to address the underlying reasons for under-vaccination in lower socio-economic groups? Are mandates minimising, or contributing to inequities in vaccination outcomes? Suggest to include a brief discussion along those lines.

Please consider adding in the conclusions a statement regarding the importance and appropriateness of strategies other than mandates for increasing vaccine uptake along the social gradient. Authors mention strategies to target vaccine hesitancy (which may be more prevalent among higher socio-economic groups), but what about the strategies to address access barriers.

Light spelling and editorial check would be helpful, examples”: in Introduction “While the immunisation coverage rates required to achieve herd immunity varies for individual diseases,…”; in Discussion “Others found NJNPay to be encourage increase uptake of catch-up vaccinations”

Author Response

Reviewer 1

In abstract, it would be helpful to clearly indicate some key factors contributing to low coverage along the social gradient, not only for the higher socio-economic group (which are mentioned and are primarily motivational). This (access and practical barriers for lower socio-economic groups) is mentioned in discussion, and should be included in the abstract as well.  

The following has been added (line 25):

Vaccine refusal and hesitancy was more evident among higher income parents while practical barriers are more likely to impact lower income families.

We have not added additional explanation in the abstract due to word restrictions, however please let us know if more is required.

Authors conclude that evaluation of mandates demonstrates continued improvements in childhood vaccination coverage. While in principle this is a correct statement, it would be helpful to include qualifiers such as “small improvements”.

Thank you for this suggestion. We have added ‘small’ improvements (see line 370)

There is a lack of discussion about the appropriateness of mandates (as a strategy to increase vaccination uptake), i.e.  are mandates helping to address the underlying reasons for under-vaccination in lower socio-economic groups? Are mandates minimising, or contributing to inequities in vaccination outcomes? Suggest to include a brief discussion along those lines.

Thank you for this comment. We have added the following (line 342)

Debate about the necessity of mandates in countries with relatively high coverage centres largely around inequities associated with lack of access to early education and opportunity for parents to secure employment, both of which are more likely to impact lower socio-economic families [3, 44]. Public health policy should aim for the least restrictive measures and avoid inequity by exacerbating socioeconomic and educational disadvantage [44]. Such debate recognises the importance of mandates, however calls for greater consideration around the medical exemption process, noting the ethical and emotional pressure this process may place on physicians [44, 45].

Additional references include:

44.  Navin M.C, Danchin M. Vaccine mandates in the US and Australia: balancing benefits and burdens for children and physicians. Vaccine. 2020;38(51):8075-7.

45.  MacDonald N.E, Harmon S, Dube E, Steenbeek A, Crowcroft N, Opel D.J, et al. Mandatory infant & childhood immunization: Rationales, issues and knowledge gaps. Vaccine. 2018;36(39):5811-8.

Please consider adding in the conclusions a statement regarding the importance and appropriateness of strategies other than mandates for increasing vaccine uptake along the social gradient. Authors mention strategies to target vaccine hesitancy (which may be more prevalent among higher socio-economic groups), but what about the strategies to address access barriers.

Thank you for this comment. We have added the following:

Continued improvements to ensure physical barriers to access are removed for lower income parents are required.

Light spelling and editorial check would be helpful, examples”: in Introduction “While the immunisation coverage rates required to achieve herd immunity varies for individual diseases,…”; in Discussion “Others found NJNPay to be encourage increase uptake of catch-up vaccinations”

Thank you for these suggestions. These edits have been made. In addition the paper has been edited throughout.

Reviewer 2 Report

This is an interesting scoping review that evaluates the impact of legislations in Australia on immunisation coverage. The authors have shown that legislations can positively impact  immunisation coverage. The methodology is clear,, and the entire paper is well written. I have spotted just two minor comments.

- the content on lines 215-219 appears to be contracdictory. Will be great if the authors can clarify this. 

-some minor improvements in punctuations are needed to improve flow. the authors will need to use :, ; and , appropriately.

Again, well done for an excellent job.

Author Response

Reviewer 2

the content on lines 215-219 appears to be contradictory. Will be great if the authors can clarify this. 

The data are describing findings from two different States in Australia – Western Australian and NSW. There is information about the different states and territories and the differing times of legislation implementation in Australia from line 90 which may set the context for the different data Nationally and from different States. Please let us know if you require additional clarification.

some minor improvements in punctuations are needed to improve flow. the authors will need to use :, ; and , appropriately.

Thank you. We have edited the paper throughout.
